# Investigation and Application of Magnetic Properties of Ultra-Thin Grain-Oriented Silicon Steel Sheets under Multi-Physical Field Coupling

**DOI:** 10.3390/ma15238522

**Published:** 2022-11-29

**Authors:** Zhiye Li, Yuechao Ma, Anrui Hu, Lubin Zeng, Shibo Xu, Ruilin Pei

**Affiliations:** 1School of Electrical Engineering, Shenyang University of Technology, Shenyang 110807, China; 2Suzhou Inn-Mag New Energy Ltd., Suzhou 215000, China

**Keywords:** grain-oriented silicon steel, magnetization angle, multi-physical field coupling, drive motor, torque density, flux density, core loss

## Abstract

Nowadays, energy shortages and environmental pollution have received a lot of attention, which makes the electrification of transportation systems an inevitable trend. As the core part of an electrical driving system, the electrical machine faces the extreme challenge of keeping high power density and high efficiency output under complex workin g conditions. The development and research of new soft magnetic materials has an important impact to solve the current bottleneck problems of electrical machines. In this paper, the variation trend of magnetic properties of ultra-thin grain-oriented silicon steel electrical steel (GOES) under thermal-mechanical-electric-magnetic fields is studied, and the possibility of its application in motors is explored. The magnetic properties of grain-oriented silicon steel samples under different conditions were measured by the Epstein frame method and self-built multi-physical field device. It is verified that the magnetic properties of grain-oriented silicon steel selected within 30° magnetization deviation angle are better than non-grain-oriented silicon steel. The magnetic properties of the same ultra-thin grain-oriented silicon steel as ordinary non-oriented silicon steel deteriorate with the increase in frequency. Different from conventional non-grain-oriented silicon steel, its magnetic properties will deteriorate with the increase in temperature. Under the stress of 30 Mpa, the magnetic properties of the grain-oriented silicon steel are the best; under the coupling of multiple physical fields, the change trend of magnetic properties of grain-oriented silicon steel is similar to that of single physical field, but the specific quantitative values are different. Furthermore, the application of grain-oriented silicon steel in interior permanent magnet synchronous motor (IPM) for electric vehicles is explored. Through a precise oriented silicon steel motor model, it is proved that the magnetic flux density of stator teeth increases by 2.2%, the electromagnetic torque of motor increases by 2.18%, and the peak efficiency increases by 1% after using grain-oriented silicon steel. In this paper, through the investigation of the characteristics of grain-oriented silicon steel, it is preliminarily verified that grain-oriented silicon steel has a great application prospect in the drive motor (IPM) of electric vehicles, and it is an effective means to break the bottleneck of current motor design.

## 1. Introduction

At present, after more than two hundred years of development, electrical machine theory has reached a very mature stage. It is an effective method to use new soft magnetic materials to break through the bottleneck of electrical machine performance [1]. Grain-oriented silicon steel is widely used in transformers and large electrical machines. In recent years, with the strict requirements for the performance of the drive electrical machine, the application of grain-oriented silicon steel in the drive electrical machine has gradually developed [2,3,4,5]. In the axial-flux electrical machine, an electrical machine hub with axial flux is designed in the literature [6], which adopts yokeless and segmented armature (YASA) structure. In order to improve the magnetic flux density of stator tooth, the stator adopts grain-oriented silicon steel, which makes the motor have higher torque density, higher efficiency and larger torque at low speed, and the effect is remarkable. In order to alleviate the problem of high iron loss during overload operation, the axial laminated flux-switching permanent magnet electrical machine (ALFSPMM) is designed with grain-oriented silicon steel, and its excellent magnetic property in the rolling direction is used to improve its torque characteristics. However, this type of motor experiences difficulties in the assembly process [7,8]. For yokeless axial flux motors, the anisotropy of grain-oriented silicon steel can be used to improve the electrical machine performance, and the output torque is increased by 3%, meanwhile, the iron consumption is reduced by 10% [9]. In other reluctance electrical machines, the high permeability of grain-oriented silicon steel along the rolling direction is also used to match the torque characteristics of the reluctance electrical machine, and both peak torque and constant torque are improved by the stator tooth splicing method [3].

Radial flux electrical machines also favor using grain-oriented silicon steel, which is applied to stator teeth in a unique splicing method. In literature [10], the stator adopts splicing layered design, which divided each layer into six sections. Adjacent layers have a different rolling direction, and they were laminated interlaced. The magnetic field distribution and the electrical machine short circuit test show that its overall loss is reduced, but the electrical machine’s performance still can be improved. In reference [11,12], orientated silicon steel is applied to the stator core and compared with the prototype by adopting five different electrical machine topologies. According to the simulation results, the distortion rate of the electrical machine back electromotive force decreases by 0.6% at least, the output torque increases by more than 2.3%, and the total stator losses decreases by 54.2% after using the grain-oriented silicon steel. The weight reduction is realized under the same output torque. In the fractional slot concentrated winding electrical machine, the segmented grain-oriented silicon steel stator structure is used to provide a more accurate modeling method to analyze the essential reasons for the torque improvement. By comparing the two-dimensional finite element simulation with the prototype, it is found that the electromagnetic torque under the same peak current is increased by 4%, and the torque fluctuation is significantly reduced from 1.6% to 0.6%. From the change in magnetic circuit parameters, the parasitic air gap generated by splicing has a great influence on the magnetic circuit of the motor. The current research is still in the simulation stage [13,14,15]. The hybrid steel splicing method was applied to traction electrical machine in literature [16,17]. Based on the comparison of the performance of grain-oriented silicon steel and non-grain-oriented silicon steel in the electrical machine, it was analyzed that under the same conditions, the maximum output torque of stator core with grain-oriented silicon steel splicing could reach 195 Nm (4.3% increase), and the rated efficiency could be increased by 2.7%. The peak efficiency can be increased by 1.5%. Similarly, in literature [18], the good performance of grain-oriented silicon steel splicing electrical machine was verified based on the grain characteristics of grain-oriented silicon steel with clear grain boundaries observed in the metallographic experiment and the comparative analysis of thermal simulation. With the gradual deepening of research, literature [19,20] began to study the influence of yoke splicing shapes on electrical machine performance, and pointed out that circular splicing was an optimal scheme, which reduced electrical machine iron loss by 36.5%. The unique Epstein measurement structure was used to separate and calculate the loss in the stator core, and the effects of different stitching shapes, stitching angles and the depth of the yoke on the performance of the electrical machine were further studied. The results show that using splicing teeth can significantly improve the torque under heavy load condition, but the modeling method still needs to be improved. In terms of finite element modeling, the accurate acquisition of material magnetic property parameters helps to improve the modeling accuracy, which, combined with the mathematical model, can effectively predict the magnetic properties of the interior permanent magnet motor, but failed when applied to the grain-oriented silicon steel motor [21].

Therefore, it is very important to accurately model and simulate the actual operating conditions of the electrical machine by using grain-oriented silicon steel in stator yoke splicing. In this paper, in order to analyze the possibility of application of grain-oriented silicon steel in IPM more accurately, the magnetic performance measurement under the coupling of thermal-mechanical-electric-magnetic multi-physical field and the performance comparison of two kinds of soft magnetic materials at different angles are carried out. In addition, a variety of properties of grain-oriented silicon steel motors are analyzed by using more refined multi-angle modeling.

## 2. Materials and Methods

Compared with conventional non-grain-oriented electrical steels, grain-oriented electrical steels exhibit excellent magnetic properties with low iron loss and high saturation flux density along the rolling direction (RD), but the magnetic properties deteriorate gradually as the magnetization direction deviates. In this paper, the properties of 0.08 mm grain-oriented silicon steel (GO) H-80 provided by Huaci Technologies Co., Ltd. (Shenzhen, China) and 0.3 mm non-grain-oriented silicon steel (NGO) provided by Angang Steel Company Limited (Anshan, China) are investigated in the background of electric vehicle drive motor application. Because of the complex service conditions, a multi-physical field non-standard ring sample method was designed to measure the magnetic properties of grain-oriented silicon steel under thermal-mechanical-electric-magnetic multi-physical fields. The magnetic properties of grain-oriented silicon steel at different angles were tested by Epstein’s square circle method. The response of grain-oriented silicon steel to external physical fields was explored by comparison and analysis, and its variation law was revealed and applied to the motor.

### 2.1. Non-Standard Ring Sample Method

As shown in Figure 1, we present a schematic diagram of the whole system for the measurement of magnetic properties under single field action and thermal-mechanical-electric-magnetic multi-field coupling by non-standard ring sample method. Among them, Figure 1b shows the test sample of grain-oriented silicon steel. The primary and secondary windings both require 200 turns. The equipment in Figure 1a can apply different temperatures through the insulation thermostat device, and the sample can be fixed on the fixture to achieve stress loading. By connecting the primary and secondary winding to the variable frequency electromagnetic excitation source, the sample can obtain a specific frequency electromagnetic field. The device can realize single physical field loading and multi-physics coupling loading. Through this device, the material response state under the actual operating conditions of the motor under the multi-physical field coupling is simulated, and the magnetic performance change in the material under the multi-physical field coupling condition is explored, which provides a strong guiding basis for the optimal design of the subsequent motor. The thermal insulation thermostat can realize the temperature field change of −75~200 °C, and the fixture can apply 0~200 Mpa stress to the sample. Combined with the excitation current at different frequencies, the magnetic characteristics measurement under the coupling of thermal-mechanical-electric-magnetic four fields can be realized.

### 2.2. Epstein Frame Method

In order to obtain the accurate magnetic properties of grain-oriented silicon steel with different magnetization angles, we use the Epstein square circle method specified by the International Electrotechnical Commission (IEC). The test conditions were 20 °C, 0 Mpa laboratory environment, and the magnetization characteristics of grain-oriented silicon steel at different magnetization angles were measured by the Epstein square circle method. As shown in Figure 2a, the Epstein frame measuring instrument consists of a primary winding, a secondary winding and a specimen as a core, which as a whole form a no-load transformer. The skeleton supporting the four coil windings consists of a hard insulating material. To verify the magnetization characteristics of grain-oriented silicon steel with different magnetization angles, the specimen is deviated from the rolling direction by a certain angle, and the shearing method is shown in Figure 2b, with α representing the different angles made with the shearing direction and the rolling direction. Taking the rolling direction, i.e., the magnetization direction or the deviation angle of 0° as an example, a group of specimens were measured through a set of coils in parallel direction, whose rolling direction was parallel to the magnetic circuit direction, and another group of specimens were perpendicular to the rolling direction. The final plotted magnetization and loss curves of the grain-oriented silicon steel at different angles is shown below.

### 2.3. Segmented Modeling of Grain-Oriented Silicon Steel Electrical Machine

The current mainstream commercial finite element simulation software uses the magnetic permeability tensor method to model grain-oriented silicon steel. In the 2D simulation, an elliptical model is constructed as the magnetic field anisotropy curve of grain-oriented silicon steel by obtaining the rolling direction and transverse magnetization characteristic curves and using interpolation fitting. This method defaults to the transverse direction as the worst magnetization direction of the magnetic field of the grain-oriented silicon steel, however, it is known from the basic material science principle of the grain-oriented silicon steel that this method has a large error. In the 3D simulation calculation, the magnetic permeability of the normal direction is obtained on the basis of 2D to form the magnetic permeability matrix, and then mathematical interpolation is performed to calculate the mathematical model as follows [22]:(1)$$B=\left[\begin{array}{ccc} \mu_{x} & 0 & 0\\ 0 & \mu_{y} & 0\\ 0 & 0 & \mu_{z} \end{array}\right]\times H$$
where *H* is the magnetic field intensity, *B* is the magnetic flux density, *μ_x_*, *μ_y_*, *μ_z_* are the permeability of the rolling direction, transverse direction and normal direction, respectively.

In view of the inaccuracy of the current mathematical model of grain-oriented silicon steel, which will lead to the calculation error of the core and the poor guidance to the actual production, this paper uses the experimental data of the magnetic properties of grain-oriented silicon steel under multiple magnetization angles in the laboratory, and uses a splicing modeling method to improve the modeling accuracy of grain-oriented silicon steel segmented motor [13,23,24]. Because the stator uses a modular grain-oriented silicon steel punching sheet, the magnetic properties of grain-oriented silicon steel at different magnetization angles are very different. In order to reflect the material properties of different regions of the stator core more accurately, a segmented modeling method is used as shown in Figure 3. Among them, T0 is parallel to the rolling direction, and its magnetization angle is 0°. In this paper, the motor model is 48 slots and eight poles, and six slots form a module. The punching and installation of the stator core are shown in Figure 4. There is a difference of 7.5° between adjacent teeth. The magnetization angle of T_1_ is 7.5°, and the magnetization angles of T_2_ and T_3_ are 15° and 22.5°. Referring to the modeling literature of grain-oriented silicon steel motor [13], the yoke magnetization angle is calculated by the average value of the inflow angle and outflow angle of the magnetic field line. The difference between Y_1_ and T_0_ is 90°, and the difference between Y_1_ and T_1_ is 82.5°. Therefore, the magnetization angle of Y_1_ is 86.25°, and the magnetization angles of Y_2_ and Y_3_ are 78.75° and 71.25°, respectively.

## 3. Results

### 3.1. Test Results of Non-Standard Ring Sample Method

#### 3.1.1. Magnetic Characteristics of Grain-Oriented Silicon Steel under Temperature Field

Drive motors are located in transportation carriers such as electric vehicles, where the installation space is small and the ambient temperature is high, in addition to the electrical machine itself is a heat source. So, the effect of temperature on the magnetic properties of the core must be accurately predicted. The flux density and loss variation in grain-oriented silicon steel under different temperature is shown in Figure 5. The saturation magnetic flux density decreases significantly with the change in temperature in the test range, and the loss increases with a rising temperature. As can be seen in Figure 5a, the initial magnetic permeability is less affected by temperature, and the degradation of permeability accelerates with the enhancement of the magnetic field; in Figure 5b, it can be seen that, contrary to the pattern of ordinary non-grain-oriented silicon steel affected by temperature, the loss of ultra-thin grain-oriented silicon steel increases with the increase in temperature.

The magnetization process is that the magnetic moment of each magnetic domain formed by spontaneous magnetization in ferromagnetic materials is transferred to the direction of the external magnetic field or close to the direction of the external magnetic field by adding an external magnetic field, which shows the process of magnetism. Therefore, this process is also the change process of magnetic domain structure under the action of external magnetic field. The change in magnetic domain structure is carried out by two magnetization modes: domain wall movement and spontaneous magnetization vector rotation (domain rotation) in the magnetic domain. It will be subjected to resistance during domain wall movement or magnetic domain rotation. The source of this resistance includes thermal strain, so as the temperature increases, the resistance of the magnetic domain becomes larger, resulting in a decrease in the saturation magnetic flux density. The classical core loss model includes hysteresis loss and eddy current loss. The hysteresis loss is related to the grain size and is less affected by temperature. The eddy current loss is related to the resistivity. As the temperature increases, the resistivity becomes larger, and the eddy current loss becomes smaller theoretically. This is the reason the core loss of ordinary non-grain-oriented silicon steel decreases with increasing temperature. From Figure 5b, it can be found that the classical loss model is no longer applicable, so the core loss model needs to be corrected to:(2)$$P_{core-loss}=P_{h}+P_{e}+P_{a}$$

Among them, *P_core-loss_* is core loss, *P_h_* is hysteresis loss, *P_e_* is eddy current loss, *P_a_* is anomalous loss. Experiments show that for grain-oriented silicon steel, the increase in temperature increases the sum of abnormal loss and eddy current loss, so the core loss increases overall.

#### 3.1.2. Magnetic Characteristics of Grain-Oriented Silicon Steel under Stress Field

When the electrical machine is operating, the stator is fed with a three-phase alternating current to produce a rotating magnetic field, and the rotor is embedded in a permanent magnet to produce an excitation field. The forces are mutual and the stator core is subjected to the same magnitude of tangential force when the rotor is rotating. In addition, the magnetic field is a space vector, so there are also radial electromagnetic and axial forces, and there are often stresses between the stator and the casing. It is known from the book [25] that stress affects the magnetization behavior of the magnetic domains, so in Figure 6, we tested the effect of stress on the magnetic flux density and core loss. It should be noted here that only tensile stresses were tested in this experiment, compressive stresses and more accurate testing methods will continue to be studied by the research team in the future.

In Figure 6a, it can be seen that compared with the 0 Mpa stress-free condition, with the increase in tensile stress, the saturated magnetic flux density of grain-oriented silicon steel increases first and then decreases, and it can also be found in Figure 6b that the core loss decreases first and then increases with the increase in tensile stress. This trend can be more intuitively reflected in three-dimensional surface diagram. This change rule is based on the principle of materials science. When ferromagnetic materials are magnetized, elastic stress is generated due to magnetostriction. When it is magnetized to elongate and is limited to not elongate, pressure is generated inside; otherwise, it produces tension. If the material is subjected to external stress or internal stress at the same time, there is an energy coupled by magneto strictive *λ_s_* and stress σ in the material. This energy is called magnetoelastic energy *E_σ_*, which is related to the size and direction of *λ_s_* and stress σ. For a certain magnet of *λ_s_*, under the action of uniaxial stress σ, when the angle between stress and magnetization is θ, the magnetoelastic energy:(3)$$E_{\sigma }=\frac{3}{2}\lambda_{s}\sigma \sin\theta$$

For materials with *λ_s_* > 0, such as Fe-Si alloy such as silicon steel, *E_σ_* is the lowest when *θ* = 0° under tension (*σ* > 0). At this time, the direction of magnetization turns to the direction of tension, which promotes magnetization. Under pressure (*σ* < 0), *E_σ_* is the lowest when *θ* = 90°. At this time, the direction of magnetization turns perpendicular to the pressure, hindering magnetization. Even if the tensile stress promotes magnetization, the magnetic domain structure of the material is fixed and cannot be stretched indefinitely, so there is an optimal tensile stress point.

This rule is exciting, because in the application process, we can carry out core stress analysis, select the best stress working point of the material, and fully develop the advantages of grain-oriented silicon steel.

#### 3.1.3. Magnetic Properties of Grain-Oriented Silicon Steels at High Frequencies

With the higher speed of the drive motor, the frequency of the electrical machine is getting higher and higher, so the characteristics of the grain-oriented silicon steel under variable frequency conditions are necessary to be investigated. The experimental results are shown in Figure 7, where it can be observed that the saturation flux density of the grain-oriented silicon steel increases with increasing frequency and the losses show an increasing trend, but the growth is slower at lower frequencies and increases faster at higher frequencies, which is also in accordance with Equation (4) [26]. It can be seen that the core loss has a certain exponential relationship with the magnetic density and frequency, so the higher the frequency is, the faster the loss increases. It is worth noting that Formula (4) considers the core loss superposition of two magnetization angles of grain-oriented silicon steel. In fact, the magnetic properties of grain-oriented silicon steel in different directions are different, so a more accurate iron loss model will be a future research direction.



(4)
$$P_{core-loss}=P_{h}+P_{e}+P_{a}=\overset{N}{\underset{i=1}{\sum }}\left\{\begin{array}{c} \left(K_{h_{RD}}\cdot B_{RD}^{1.6}+K_{h_{TD}}\cdot B_{TD}^{1.6}\right)\cdot f^{1}\\ \left(K_{e_{RD}}\cdot B_{RD}^{2}+K_{e_{TD}}\cdot B_{TD}^{2}\right)\cdot f^{2}\\ P_{a} \end{array}\right.$$



Among them, *N* is the number of core segments, *K_hRD_*, *K_hTD_*, *K_eRD_* and *K_eTD_* are the hysteresis loss coefficient in the rolling direction, the transverse hysteresis loss coefficient, the eddy current loss coefficient in the rolling direction and the transverse eddy current loss coefficient, respectively. They are related to the material properties and can be obtained by fitting the material test data.

#### 3.1.4. Magnetic Characteristics of Grain-Oriented Silicon Steel under Multi-Physical Field Coupling

The operating environment of the electrical machine is in a multi-field coupled space such as thermal-force-electrical-magnetic, etc. In order to achieve the accuracy of the electrical machine simulation model, the realistic characterization of the silicon steel material in multi-physics fields is especially important [27]. Under the multi-physics test platform built in the laboratory, the team measured the loss variation pattern and the flux density of 1000 A/m at 50 Hz for 400 Hz and 1 T ultra-thin grain-oriented silicon steel under multi-field coupling. From Figure 8a, it can be seen that with the increase in stress at a certain temperature, the core loss of grain-oriented silicon steel decreases and then increases, and at about 30 Mpa, the loss is the smallest; with the increase in temperature at a certain stress, the core loss of grain-oriented silicon steel increases slightly, and the higher the temperature, the less obvious the change trend. In Figure 8b, the flux density increases and then decreases with the increase in stress at a certain temperature, and the flux density is maximum at about 35 Mpa; the flux density shows a decreasing trend with the increase in temperature at a certain stress. Comparing the magnetic properties of grain-oriented silicon steel under single physical and multi-physical field coupling conditions, it can be found that the change rules of the two are similar, but the specific quantity changes are still different, which is the influence of multi-physical field coupling. We will continue to study the coupling mechanism in depth.

### 3.2. Experimental Results of Epstein Frame Method

#### 3.2.1. Magnetic Characteristics of Grain-Oriented Silicon Steel at Different Angles

The Epstein frame is one of the most accurate methods to test the magnetic properties of silicon steel. In order to understand the variation in the magnetic properties of the experimental grain-oriented silicon steel from the rolling direction, we tested the magnetic properties of the samples in the range of 0–90°. Figure 9 shows some experimental results. It should be noted that the magnetic properties of a grain-oriented silicon steel within 0–360° can be determined only by measuring 0–90°, because 90–180° and 0–90° are symmetrical, and 180° is a change period. Figure 9a shows the change in saturation magnetic induction intensity with increasing deviation angle. It can be seen that the performance of grain-oriented silicon steel decreases seriously with the increase in deviation angle, and the deterioration is more serious after 30°. In addition, it can be clearly seen from the local amplification diagram that the permeability decreases with the increase in the deviation angle, which is the reason for the change in its macroscopic characterization (magnetization curve and iron loss curve); Figure 9b shows the loss change, which can be found to be continuously deteriorating, and the deterioration is also serious after 30°.

The ultra-thin grain-oriented silicon steel studied is Si-Fe alloy, and the crystal is a cubic structure. According to the principle of metal materials, [100] that is, the edge length direction is the easy magnetization direction; [111] and the body diagonal direction is the difficult magnetization direction. This magnetic difference in different crystal axis directions is called magneto crystalline anisotropy. The reason for magneto crystalline anisotropy is that the regular arrangement of atoms or ions in the crystal causes an inhomogeneous electrostatic field with spatial periodic changes, which changes the orbital angular momentum of electrons in the atom, but its average change may be zero. So, the atomic magnetic moment shown in the crystal is mainly its total electron spin magnetic moment. On one hand, the electron is affected by the non-uniform static electromagnetic field of the spatial periodic change, and at the same time, there is an exchange effect between the electron orbits of the adjacent atoms. The orbital motion of the electron is coupled with its spin. Thus, the magnetic moment has different energy levels in different directions of the crystal.

The magneto crystalline anisotropy energy *E_k_* is usually expressed as a power series of the direction cosine of the saturation magnetization vector *M_s_* relative to the main crystal axis. For a cubic crystal, due to its cubic symmetry, *E_k_* is:(5)$$E_{k}=K_{0}+K_{1}\left(\alpha_{1}^{2}\alpha_{2}^{2}+\alpha_{2}^{2}\alpha_{3}^{2}+\alpha_{3}^{2}\alpha_{1}^{2}\right)+K_{1}\alpha_{1}^{2}\alpha_{2}^{2}\alpha_{3}^{2}+\cdots$$
where *α*_1_, *α*_2_ and *α*_3_ are direction cosines of Ms relative to three [100] axes; *K*_0_, *K*_1_ and *K*_2_ are the crystal anisotropy constants, which vary with the material and temperature. The sign and value of *K*_0_, *K*_1_ and *K*_2_ determine the easy magnetization direction and difficult magnetization degree of the material. Since *K*_0_ is independent of the magnetization direction, and in most cases, *K*_2_ is too small to be ignored, *E_k_* is mainly represented by *K*_1_. Grain-oriented silicon steel (this material) *K*_1_ > 0, [100] the direction of the lowest energy, so this direction is the easy magnetization direction. According to Equation (5), the change trend of *E_k_* with the magnetization angle can be achieved, and *E_k_* is the lowest in the 0° direction parallel to the rolling direction in the magnetization direction. At this time, the saturation flux density is the largest, the core loss is the lowest, and the permeability is also the highest. The calculation of *E_k_* can explain the experimental results in Figure 9. With the deviation of magnetization angle, the magnetic properties of grain-oriented silicon steel gradually deteriorate. In the direction of about 55°, the *E_k_* is the largest, the corresponding permeability is the lowest, and the magnetic flux density is the lowest at the magnetic field strength of 4000 A/m. The multi-angle test results of grain-oriented silicon steel are consistent with the calculation results of Formula (5).

In the previous paragraphs, the effect of different frequencies on grain-oriented silicon steel was discussed in the non-standard ring method, and it is impossible to compare the effect of different frequencies at different angles. Therefore, the effect of frequency on the magnetic properties of ultra-thin grain-oriented silicon steel at multiple angles was further measured using the Epstein frame method, and the results are shown in Figure 10. The amount of data is large, and the results are only listed for the cases of 50 Hz and 400 Hz according to the application equipment of this grain-oriented silicon steel. It is not difficult to find that the variation pattern in Figure 10a under different deviation angles is the same as that in Figure 9a, as the magnetization angle increases, the saturation magnetic flux density decreases. The magnetic flux density is basically the same at both frequencies. In Figure 10b it is obvious that the losses are affected by the frequency and show an order of magnitude increase, much larger than the effect of the different deviation angles.

#### 3.2.2. Magnetic Flux Density Variation Diagram of Two Materials at Different Angles

Finally, by performing the experiments, the variation in magnetic flux density of grain-oriented silicon steel and non-grain-oriented silicon steel with deviation angle α within 180 ° was compared under the condition of 50 Hz and 4000 A/m. As shown in Figure 11, the magnetic flux density of the ultra-thin grain-oriented silicon steel selected in the experiment is higher than that of the non- grain-oriented silicon steel within 30°. Therefore, using the grain-oriented silicon steel to replace the non- grain-oriented silicon steel within 30° will bring obvious advantages to the motor core. It should be emphasized here that the advantages of different grain-oriented silicon steel angle (grain-oriented silicon steel magnetic properties better than the same level of non-grain-oriented silicon steel magnetic properties of the angle) is different, according to the laboratory ‘s previous range of 20~30°.

### 3.3. Drive Electrical Machine Based on Grain-Oriented Silicon Steel

The reference model for the design of the grain-oriented silicon steel electrical machine is a 48-slot, eight-pole drive electrical machine for electric vehicles, which has the basic performance parameters required, as shown in Table 1.

The conventional non-grain-oriented silicon steel electrical machine cannot meet the critical performance requirements of this drive electrical machine, so this paper attempts to use ultra-thin grain-oriented silicon steel in the stator core. Faced with the specificity of the superior performance of grain-oriented silicon steel in specific directions, the angular range of grain-oriented silicon steel superior to non-grain-oriented silicon steel was determined based on material tests, as shown in Figure 4, and the electrical machine stator was determined to be divided into six sections (all six sections are identical, using grain-oriented silicon steel punching sheets), and further finite element precision modeling of the designed electrical machine in the form of iron core splicing was performed. The detailed settings of finite element numerical calculation are shown in Table 2.

The reference electrical machine model is shown in Figure 12a. The electrical machine requires high power density and high torque density, so the non-grain-oriented silicon steel stator core is close to saturation to achieve the performance requirements under the limited constraints. According to the working principle of the electrical machine, the stator flux density is the largest at the teeth, and the magnetic flux density at the yoke is relatively low, so how to improve the saturation flux density at the teeth is the key point. According to the magnetic field distribution characteristics of the electrical machine, the investigated alternative model electrical machine stator core uses a 60° fan-shaped oriented silicon steel sheet, with the center tooth parallel to the rolling direction, i.e., the center tooth is 0° magnetization characteristic of the grain-oriented silicon steel, and the outermost layer of the whole fan-shaped sheet does not exceed 30°, which is just within the dominant range of the grain-oriented silicon steel. Applying the topological simulation model as in Figure 12b, it can be found that the magnetic flux density is increased at the same tooth position compared with Figure 12a, and it is increased by 2.20% at the center tooth.

Because the two motor models only have different stator materials and other structural parameters are exactly the same, the end effect is ignored, so only the 2D finite element numerical calculation is carried out. In the follow-up work, we are ready to manufacture the prototype, which will take into account more parameters affecting the performance of the motor, such as the parasitic air gap problem at the splicing of adjacent grain-oriented silicon steel stator teeth and the mathematical model for more accurate prediction of actual motor performance.

Figure 13 shows the torque comparison of the two motors. Compared with the non- grain-oriented silicon steel stator, the result torque of the grain-oriented silicon steel segmented stator can reach 343.38 Nm, while the non-grain-oriented silicon steel stator structure can only reach 336.05 Nm even if it approaches saturation and reaches the limit, which cannot meet the peak torque of 340 Nm. However, it cannot be ignored that the torque ripple is more than 3% in the stator structure of grain-oriented silicon steel. One of the reasons is that the splicing material modeling is adopted, which will not occur in engineering production.

The speed of an electric vehicle drive motor varies during its operation, so, in order to reflect the superiority of the segmented grain-oriented silicon steel stator more clearly and to fully reflect the electrical machine performance, a map chart comparison of torque-speed efficiency was presented. Under the condition of utilizing grain-oriented silicon steel, the range of efficiency greater than 97% is significantly enhanced, and the maximum efficiency is up to 98%, which shows the particular advantage of using grain-oriented silicon steel in drive motors.

As a matter of fact, the drive motor is in a temperature-stress-electric-magnetic multi-field coupling situation during operation; however, in the electrical machine research stage, the data of material modeling often only considers the electromagnetic field, thus causing wrong evaluation of the product. By building a multi-physics field experimental platform independently, our experimental team measured the performance of grain-oriented silicon steel under multi-field coupling and compared it with the standard Epstein frame method measurement results, the data comparison is shown in Figure 14. The multi-physics field coupling experimental conditions are 75 °C, 75 Mpa and 400 Hz, the conventional conditions are room temperature, no stress and 400 Hz, the purpose is to verify the real operation of the electrical machine conditions and the errors formed by inaccurate material modeling during the R&D phase.

In fact, the drive motor is in a temperature-stress-electric-magnetic multi-field coupling during operation, however, in the motor development stage, the material modeling data often only consider the electromagnetic field, resulting in the wrong assessment of the product. By independently building a multi-physical field experimental platform, the experimental team measured the performance of grain-oriented silicon steel under multi-field coupling, and compared it with the measurement results of the standard Epstein square ring method; see Figure 15. The metaphysical field coupling experimental conditions are 75 °C, 75 Mpa and 400 Hz, conventional conditions are room temperature, no stress and 400 Hz, the purpose is to verify the error caused by inaccurate material modeling under the real operating conditions of the motor and the research and development stage.

## 4. Conclusions

Compared with non-grain-oriented silicon steel, grain-oriented silicon steel has obvious advantages and drawbacks. This paper comprehensively explores the magnetic property variations law of an ultra-thin grain-oriented silicon steel under different physical fields and multi-field coupling conditions, and verifies its application possibilities in electric vehicle drive motors. Under the temperature field, with the increase in temperature, the magnetic induction strength decreases and the loss increases (the loss of ordinary non-grain-oriented silicon steel decreases); under the stress field, with the increase in tensile stress, the magnetic induction strength increases and then decreases, and the loss decreases and then increases, but both are better than the non-stressed state. Under the electromagnetic field of different frequencies, the magnetic induction decreases as the frequency increases, and the loss gradually deteriorates, and the higher the frequency, the more serious the deterioration; under the conditions of temperature-stress-electrical-magnetic multi-field coupling, when the stress is certain, the temperature has less effect on the loss, the magnetic induction strength decreases slightly with the increase in temperature, and the temperature is certain with the increase in stress, the performance of grain-oriented silicon steel first improves and then deteriorates, and in the 30–40 MPa range it is performs the best. The magnetic properties of the grain-oriented silicon steel are seriously affected by the deviation angle, which is similar to the literature, and it is worth mentioning that the magnetic properties of this ultra-thin grain-oriented silicon steel are better than those of the non-grain-oriented silicon steel in the range of 30°.

In terms of application, our laboratory intends to develop a spliced grain-oriented silicon steel electric electrical machine product because ordinary non-grain-oriented silicon steel cannot meet the high-performance requirements of the drive motor. In order to improve the accuracy of finite element modeling, this paper adopts a design model based on multi-angle data stitching. The conventional model is unable to meet the performance index when the flux density reaches the limit. After adopting the grain-oriented silicon steel, the flux density of the teeth is significantly improved, and then the torque is also improved by 2%, breaking the performance bottleneck, and its efficiency can reach 98%, expanding the high efficiency area nearly twice. The initial verification of the orientation of silicon steel in the drive motor has a greater application perspective, is an effective method to break the current electrical machine design bottleneck.

## Figures and Tables

**Figure 1 materials-15-08522-f001:**
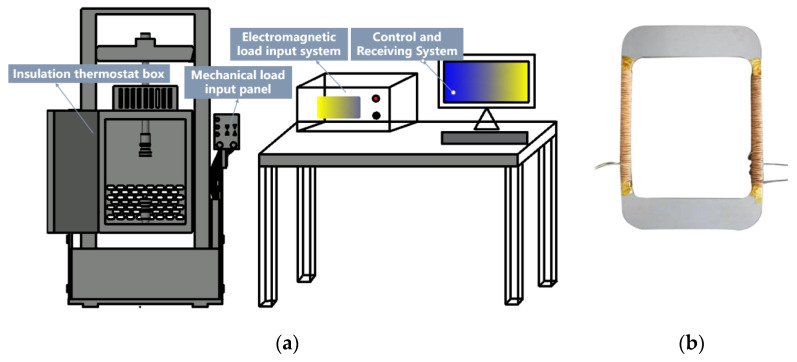
The overall schematic diagram of the multi-physics coupling test system. (**a**) Multi-physics coupling test system; (**b**) grain-oriented silicon steel stress test specimen.

**Figure 2 materials-15-08522-f002:**
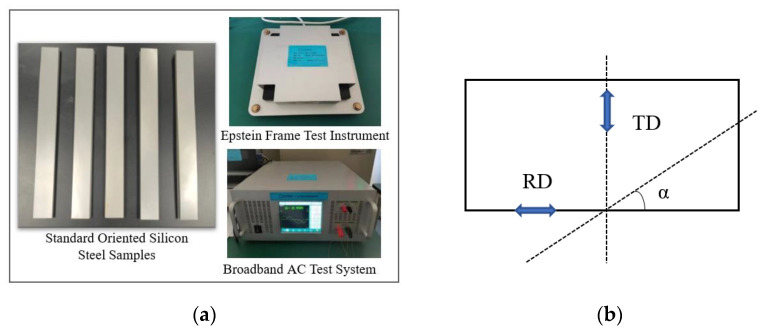
Test equipment and samples. (**a**) Overall schematic diagram of Epstein magnetic measurement system; (**b**) shearing of grain-oriented silicon steel sheet.

**Figure 3 materials-15-08522-f003:**
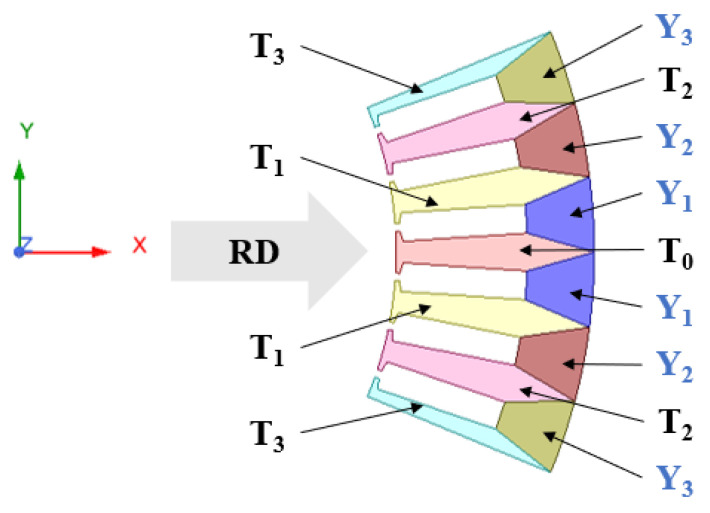
Schematic diagram of segmented modeling of grain-oriented silicon steel stator core.

**Figure 4 materials-15-08522-f004:**
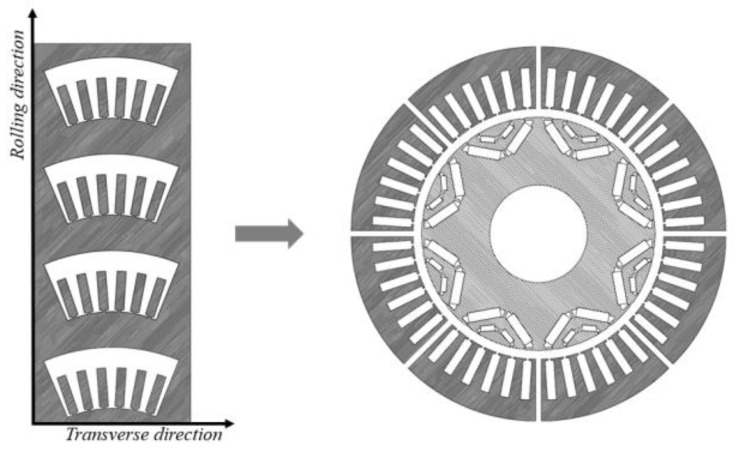
Example of GO laminations cutting to be installed in a 48 s/8 p IPM modular machine.

**Figure 5 materials-15-08522-f005:**
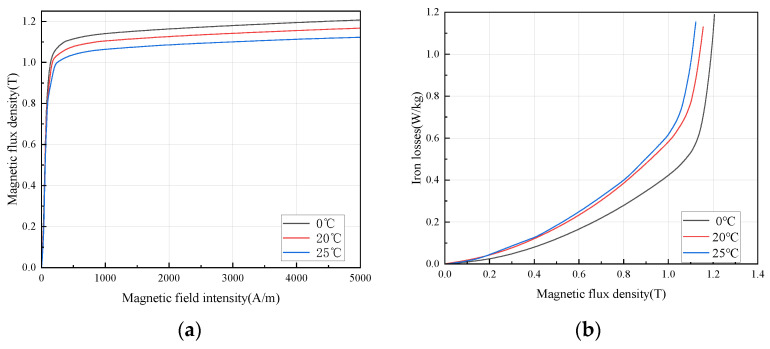
Magnetic properties of grain-oriented silicon steel at different temperature fields (**a**) magnetization curves of H-80 @ 50 Hz and 0 Mpa (**b**) iron loss curves of H-80 @50 Hz and 0 Mpa.

**Figure 6 materials-15-08522-f006:**
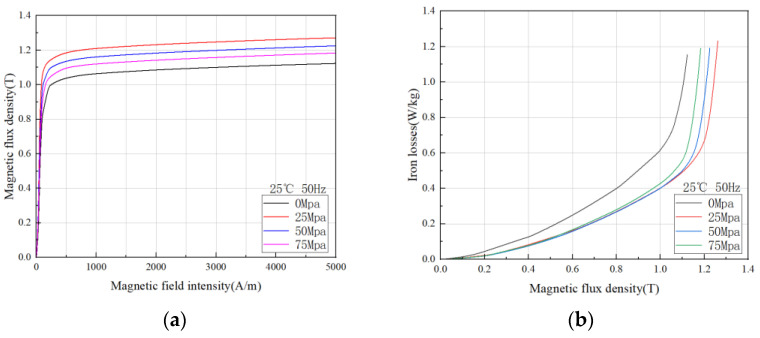
Magnetic properties of grain-oriented silicon steel under different stress fields (**a**) magnetization curves of H-80 material @25 °C and 50 Hz under different stresses; (**b**) iron loss curves of H-80 material @25 °C and 50 Hz under different stresses.

**Figure 7 materials-15-08522-f007:**
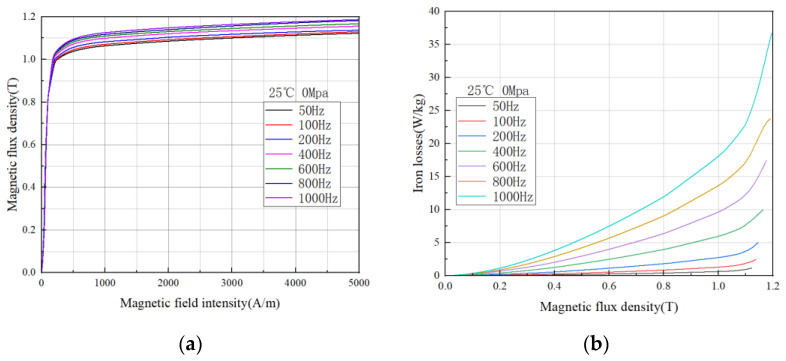
Magnetic properties of grain-oriented silicon steel at different frequencies (**a**) magnetization curves of H-80 material @25 °C and 0 Mpa (**b**) iron loss curves of H-80 material @25 °C and 0 Mpa.

**Figure 8 materials-15-08522-f008:**
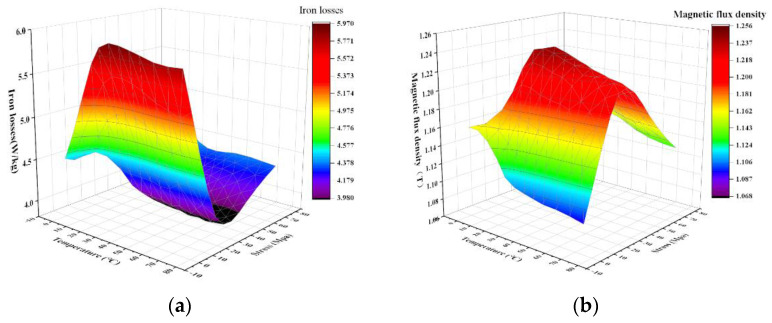
Magnetic properties of grain-oriented silicon steel under multi-physical field coupling (**a**) loss variation in H-80 under tensile stress @1 T and 400 Hz; (**b**) magnetic density variation in H-80 under tensile stress@1000 A/m and 50 Hz.

**Figure 9 materials-15-08522-f009:**
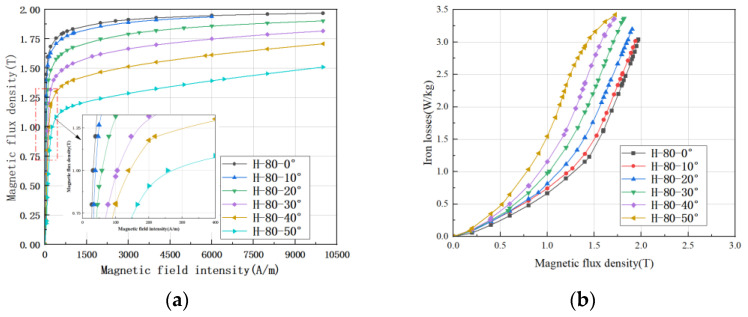
Magnetic characteristics of grain-oriented silicon steel at different angles (**a**) magnetization curves of H-80 material at different angles @50 Hz; (**b**) iron loss curves of H-80 material at different angles @50 Hz.

**Figure 10 materials-15-08522-f010:**
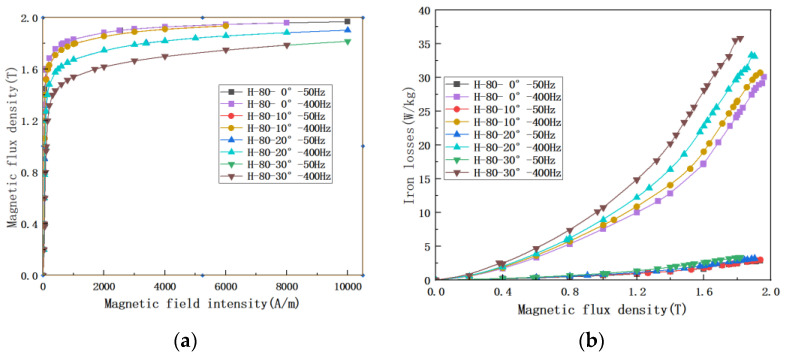
Magnetic characteristics of grain-oriented silicon steel at different angles (**a**) magnetization curves of H-80 material at different angles @400 Hz; (**b**) iron loss curves of H-80 material at different angles @400 Hz.

**Figure 11 materials-15-08522-f011:**
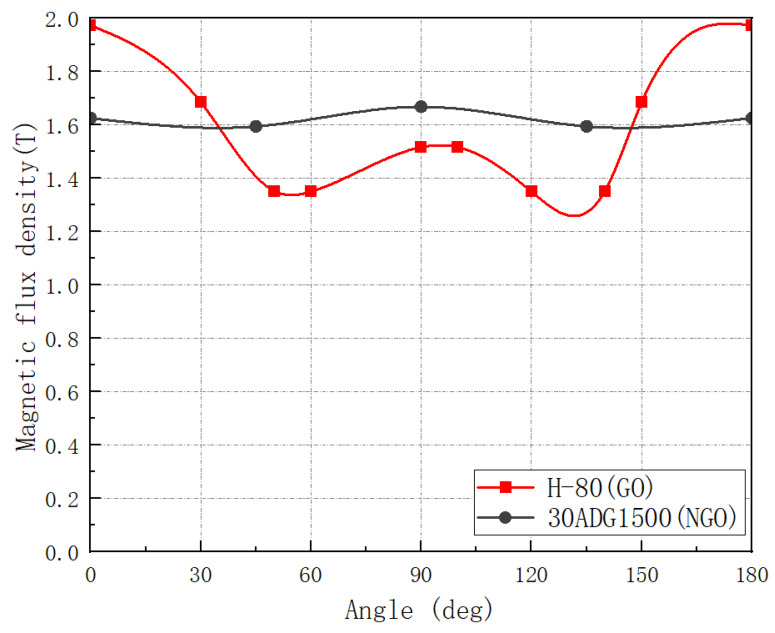
Comparison of saturation flux density of oriented and non-grain-oriented silicon steel with different magnetization angles.

**Figure 12 materials-15-08522-f012:**
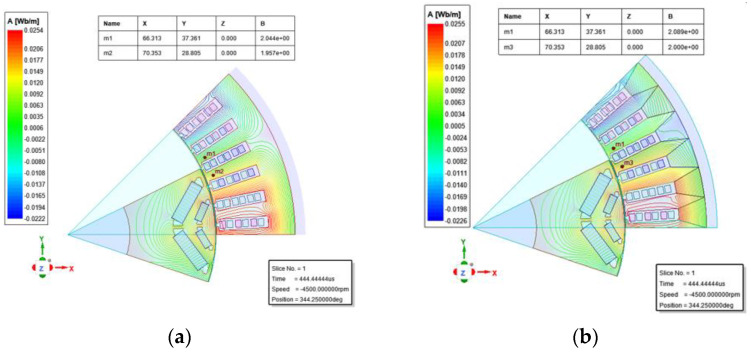
Electrical machine finite element simulation model. (**a**) Traditional electrical machine modeling reference model (NGO-IPM) (**b**) multi-angle spliced grain-oriented silicon steel electrical machine model (GO-IPM).

**Figure 13 materials-15-08522-f013:**
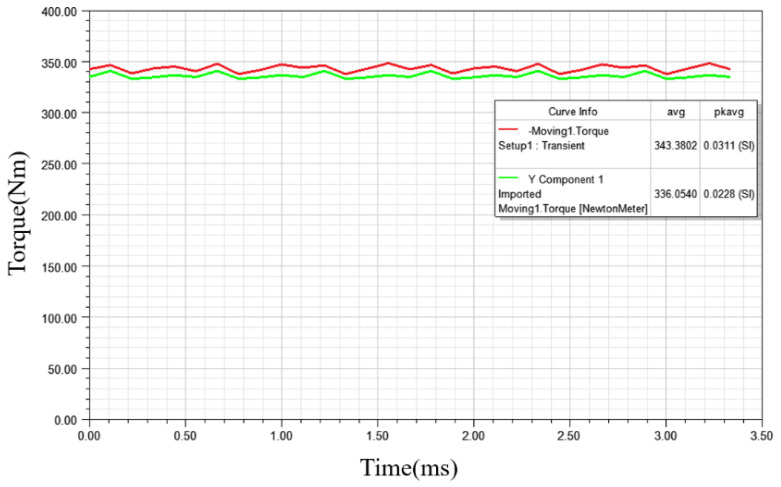
Torque comparison of non-grain-oriented silicon steel electrical machine (green line) and grain-oriented silicon steel electrical machine (red line).

**Figure 14 materials-15-08522-f014:**
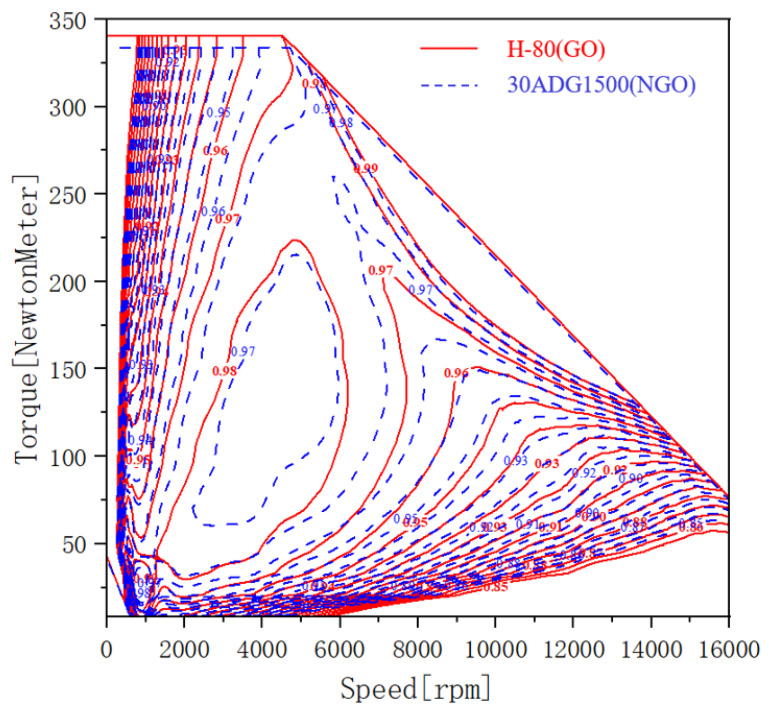
Comparison of efficiency map between traditional non-grain-oriented silicon steel motor model and grain-oriented silicon steel motor model.

**Figure 15 materials-15-08522-f015:**
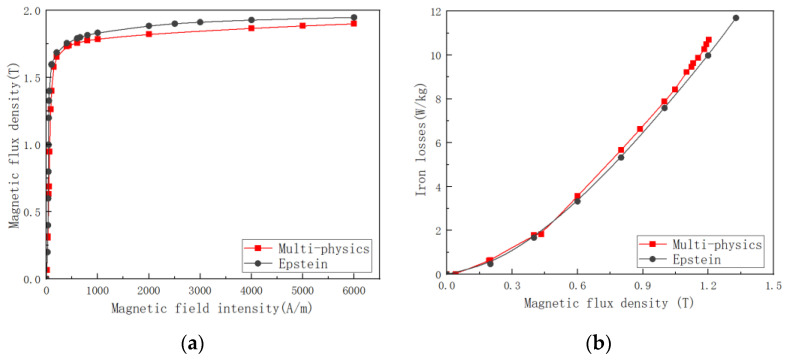
Comparison of magnetic properties of grain-oriented silicon steel under multi-physical field coupling and conventional measurements. (**a**) magnetization curves of H-80 material @75 °C, 75 Mpa and 400 Hz. (**b**) iron loss curves of H-80 material at different angles @@75 °C, 75 Mpa and 400 Hz.

**Table 1 materials-15-08522-t001:** Electrical machine performance parameters.

Performance	Parameters
The rated line voltage	350 VDC
Rated current	260 A
Peak power	≥145 kW
Speed	4500 rpm
Peak torque	340 Nm
The efficiency of	96%

**Table 2 materials-15-08522-t002:** Finite Element Numerical Calculation Detailed Settings.

	NGO-IPM	GO-IPM
Total number of elements	3668	3578
Solver Time Step	0.00011 s	0.00011 s
Solver Boundaries	Vector Potential (Value: 0)	Vector Potential (Value: 0)
	Independent	Independent
	Dependent (Bdep = −Bind)	Dependent (Bdep = −Bind)

## Data Availability

Not applicable.

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
