# Peer review of "Investigation and Application of Magnetic Properties of Ultra-Thin Grain-Oriented Silicon Steel Sheets under Multi-Physical Field Coupling"

_materials, 2022, doi:10.3390/ma15238522_

Round 1

Reviewer 1 Report

Li et al studied use of magnetic properties of ultra-thin oriented silicon steel sheets for different applications. This paper explores the subject into four fields including thermal, mechanical, electrical and magnetical field. The magnetic properties of the oriented silicon steel were better than non-oriented silicon steel at a certain angle. More interesting findings like this were reported and the paper was reviewed for a candidate paper in materials-MDPI journal. However, certain revisions are required before the paper can be accepted for publication. Some points are –

[1] Abstract and in fact whole paper is full of very long sentences. Please make them short and concise. Moreover, please make the abstract short and add few quantities of improvements properties in % in abstract.  

[2] What is “YASA structure” in introduction. Please expand. Also, please refer 4-5 more papers from MDPI in introduction and discuss the advancements of present work over existing literature in MDPI.

[3] In section-2, please provide the commercial name of the materials, company of purchase and their physical properties.

[4] Various parameters such as frequency, temperature, pressure and angle are varied and their effect of magnetic effect and other properties are represented. However, what is the best candidate for all individual parameters and why must be highlighted in abstract, results and conclusion.

Good Luck for revisions !

Reviewer 2 Report

The present manuscript is devoted to the characterization through measurement of the ultrathin-oriented silicon steel sheet (GO0.08 material), in view of its utilization in electrical machines’ stator manufacturing. The positive effect presented by the use of the above-mentioned material is assessed utilizing a 2D electromagnetic field numerical analysis software (undisclosed). The measurement part using the Epstein frame is quite a routine approach, even though it is performed at different environmental temperatures and mechanical stress values. The novelty presented by this part is difficult to identify since the measuring scheme does not present any possible improvements worth to be reported. Neither a microscopic model to match or explain at the physical level the obtained results and their corresponding variation trends is invoked or proposed. Consequently, the corresponding sections spanning from 2.1 to 3.2 present themselves more likely as a laboratory test report than a scientific paper. The 2D software simulation part is also straightforward with no obvious contribution concerning the numerical method or the device modeling method, being carried out by an EM field software. Possibly, the device’s adopted segmented structure may present more interest, but the necessary justification and clarifications related to it are cursory. How is this structure designed, which were the principles, guidelines, targeted parameters, etc. for the possibly performed optimization routine, all remain undisclosed issues? Was an optimization routine performed to obtain the structure shown in Figure 3 or just an empirical trial-and-error procedure?

Apart from the above considerations the following concerns, remarks, and corrections can be expressed:

1.   The type of the considered electrical machine is not specified (DC or AC). The considered working regime (motor or generator) is also not specified. It might be finally implicitly suggested that a DC motor is considered (see Table 1), but, earlier in the paper, due to its unspecific structural adopted solution, there are hints favoring even a synchronous machine (the PM magnets embedded into the rotor). An explicit statement concerning the above-mentioned aspects is certainly desirable or even necessary.

2.   The 2D numerical simulation of the proposed machine has undisclosed parameters: number of mesh elements (whether they are fine or coarse, and where?), utilized boundary conditions, solver precision, etc. Since the authors’ intention is to push the limits of the machine through the existing “bottleneck”, in terms of its performance (torque, efficiency, etc.), one may question if 2D modeling is sufficiently accurate to model a classical 3D structure like an electrical machine. Can the end effect be neglected (as it is in a 2D simulation) when such a narrow margin of improvement of 1-2% for the considered performance parameters is at stake? One should be aware that a 2D simulation vs. a 3D simulation will induce significant errors of 10-15 % or even higher, depending on the z-axis length of the structure compared to the other xy-plane cross-sectional dimensions.

3. Additional 39 aspects are marked on the attached PDF containing the initial manuscript. Please refer to it.

Reviewer 3 Report

The authors present research and application of magnetic properties of ultra-2 thin-oriented silicon steel sheets. The work is well constructed, but contains some inaccuracies that need to be cleared up.

1.What is the chemical composition of the oriented steel and the nonoriented steel to which the comparisons were made?

Are these differences not due to slight differences in the composition of the steel?

2 Please describe the methodology for obtaining the oriented steel and determining the direction in which it was oriented.

3. Figure 7 requires improvement, descriptions of axes and values are not legible, similarly in Figure 11- the values are not legible

Round 2

Reviewer 1 Report

Acceptable

Reviewer 2 Report

My comments and concern were satisfactorily responded to by the authors. I consider the manuscript has significantly gained in clarity and quality. I have no further remarks.